# Player load in male elite soccer: Comparisons of patterns between matches and positions

**Terje Dalen**[1]*, **Tore Kristian Aune**[1], **Geir Håvard Hjelde**[2], **Gertjan Ettema**[3],
**Øyvind Sandbakk**[3], **David McGhie**[3]

**1** Department of Physical Education and Sport Science, Nord University, Levanger, Norway, **2** Rosenborg FC, Trondheim, Norway, **3** Centre for Elite Sports Research, Department of Neuromedicine and Movement Science, Norwegian University of Science and Technology, Trondheim, Norway

* terje.dalen@nord.no

**Data Availability Statement:** All relevant data are within the paper and its Supporting Information files.

## Abstract

Our primary aim was to explore the development of player load throughout match time (i.e., the pattern) using moving 5-min windows in an elite soccer team and our secondary aim was to compare player load patterns between different positions within the same team. The dataset included domestic home matches ($n = 34$) over three seasons for a Norwegian Elite League team. Player movements (mean ± SD age 25.5 ± 4.2 years, height 183.6 ± 6.6 cm, body mass 78.9 ± 7.4 kg) were recorded at 20 Hz using body-worn sensors. Data for each variable (player load, player load per meter, total distance, accelerations, decelerations, sprint distance, high-intensity running distance) were averaged within positions in each match, converted to z-scores and averaged across all matches, yielding one time series for each variable for each position. Pattern similarity between positions was assessed with cross-correlations. Overall, we observed a distinct pattern in player load throughout match time, which also occurred in the majority of individual matches. The pattern shows peaks at regular intervals (~15 min), each followed by a period of lower load, declining until the next peak. The same pattern was evident in player load per meter. The cross-correlation analyses support the visual evidence, with correlations ranging 0.88–0.97 ($p < .001$) in all position pairs. In contrast, no specific patterns were discernible in total distance, accelerations, decelerations, sprint distance and high-intensity running distance, with cross-correlations ranging 0.65–0.89 ($p < .001$), 0.32–0.64 ($p < .005$), 0.18–0.65 ($p < .005$ in nine position pairs), 0.02–0.38 ($p < .05$ in three pairs) and 0.01–0.52 ($p < .05$ in three pairs), respectively. This study demonstrated similarity in player load patterns between both matches and positions in elite soccer competition, which could indicate a physical "pacing pattern" employed by the team.

## Introduction

For optimal performance in team sports like soccer (association football), players are required to maximize their technical, tactical, and physical abilities. The physical demands of soccer matches are characterized by a constant variation between low- (e.g., standing and walking),

**Funding:** The funder (Rosenborg FC) provided support in the form of salaries for author [G.H.H.], but did not have any additional role in the study design, data collection and analysis, decision to publish, or preparation of the manuscript.

**Competing interests:** The commercial affiliation (Rosenborg FC) does not alter our adherence to all PLOS ONE policies on sharing data and materials by including the following statement: 'This does not alter our adherence to PLOS ONE policies on sharing data and materials.'

high- (e.g., running), and very high-intensity (e.g., accelerations, decelerations and sprinting) activities [1–3]. Along with additional sport-specific activities (e.g., tackles, turns, headers, dribbles), these locomotor activities constitute the total physical load of a player during training and matches [4]. However, the total physical load of the players is determined by a combination of direct involvement in play, responding to movements of attacking players, tactical restrictions, and willingness to support team-mates [5]. These variations are likely to result in a relatively large match to match variability in physical performance [6, 7].

Time-motion analyses have provided accurate and objective quantification of the players' activities, and therefore improved our understanding of the physical demands in soccer [8–11]. However, measurements of different locomotor classifications or speed zones may be insensitive to the totality of mechanical stresses common to team sports. Tri-axial accelerometers provide complementary information to time-motion analysis for understanding player load during matches and training [12, 13] as they record the acceleration of body movement in three dimensions, which better estimates the players' physical exertion. Therefore, manufacturers of global positioning systems (GPS) and local positioning measurements (LPM) have incorporated high-resolution triaxial accelerometers as a measure of player load. Such analyses are shown useful for validly quantifying the physical demands in soccer [12, 14–16], in which various estimations of player load are regarded as acceptable measures of external load and largely correlated to players' physiological and perceptual responses to training [17, 18]

To date, monitoring external training and match load measures in soccer has tended to rely on results based on locomotor activities. In previous analyses of soccer matches, considerable heterogeneity has been observed in the within-match development of locomotor activities (total distance, HiR, sprint, accelerations and decelerations) throughout match time (i.e., the pattern) across studies [6, 9, 19–25]. Some studies report a reduction in total and high-intensity running (HiR) distances toward the end of each half [9, 26], whereas others do not find such changes [20, 21]. These contradictory results are likely caused by different measurement systems, different tactical elements, opponents' playing style, pacing strategies, score line, and team formation, which would all affect the players' ability to regulate and maintain their physical effort [22]. However, previous studies show high variability in high-speed activities within matches and that individual players show inconsistency in high-speed activity (i.e., HiR and sprinting) across matches [6, 23]. A component of soccer matches that has received relatively less attention is the players' number of accelerations and decelerations [19], although some previous studies suggest that inter- and intra-individual variability is smaller for accelerations compared to distance-related measures [6, 24]. Additionally, a recent study found a continuous reductional pattern in accelerations over the course of a match and after peak working periods of a match, which was consistent across positions [25].

In the existing literature, the within-match player load based on three-dimensional movement analyses has been investigated using a standardised soccer simulation with 15-min standardised activity blocks [27]. Here, the authors found that player load increased over time in each half, likely due to a change in movement strategy and/or a reduced locomotor efficiency [27]. In contrast to this, reductions in player load were identified in the latter stages of each half in the analyses of 86 matches in U-21 English Championship teams [14]. However, in the same 15-min time periods, the player load per total distance covered increased, suggesting an increased loading for every given meter covered on the pitch [14]. These investigations have allowed a general determination of player load patterns during soccer matches and soccer-specific intermittent exercises. However, to understand more in detail how teams and individual players distribute their player load and related locomotor activities throughout soccer matches, the same factors need to be analyzed over shorter time-periods than 15-min blocks. More instantaneous analyses of player load and the corresponding activities during soccer matches

would logically show a variable "pacing" influenced by e.g., tactical elements, player position, and the level of the opponents. In order to quantify this across e.g., positions, the similarity of patterns throughout the duration of matches must be analyzed. Long term analyses of such data and the relationships to changes in tactics, different opponents, and match outcome have the potential to provide imperative understanding of how the team and the players in different positions distribute the load (i.e., "pacing strategies") during different types of matches.

Since analyses based on predefined periods cannot provide information about the "real" peaks and valleys in the analysis of patterns throughout a match, moving windows is a potential solution, providing more accurate information about player load and locomotor variables (total distance, acceleration, deceleration, HiR and sprint). Our primary aim was to explore the patterns of player load, as well as locomotor variables for comparison, with analyses from moving 5-min windows in an elite soccer team. Our secondary aim was to compare these patterns between different positions within the same team.

## Methods

### Participants

The dataset includes domestic home matches ($n = 34$) over three full seasons for a team in the Norwegian Elite League. In one of the seasons, the team participated in the Europe League group stages. All matches were played on a grass surface. Movements of all players (mean ± SD age 25.5 ± 4.2 years, height 183.6 ± 6.6 cm, body mass 78.9 ± 7.4 kg) were observed, and only data from the 39 players completing an entire match were used ($n = 212$: complete match data of players, goalkeepers excluded). The sample included eight central defenders (CD, $n = 47$), six external defenders (ED, $n = 52$), six central midfielders (CM, $n = 46$), 11 external midfielders (EM, $n = 40$), and eight attackers (ATT, $n = 27$). Some players participated in different positions across, but not within, the matches included in the data material. Following an explanation of the procedures, all participants gave verbal and written informed consent to participate in the study. The study was conducted according to the Declaration of Helsinki and has been approved by the Norwegian Social Science Data Services (reference number 468065).

### Study design and methodology

This study used a fully automatic sport tracking system to evaluate match performances of professional soccer players at the elite level over three full seasons. Player movement was captured by small, body-worn sensors located at the lumbar region, continuously recording the players' actions. Data were transferred by microwave radio channel to 10 RadioEye™ sensors (ZXY SportTracking, ChyronHego, Trondheim, Norway) mounted in the team's home arena. Player movement was registered at 20 Hz. Accelerations and decelerations were recorded when they reached limits of 2 m·s$^{-2}$ and -2 m·s$^{-2}$, respectively, and a HiR category of >19.8 km·h$^{-1}$ and sprint category of >25.2 km·h$^{-1}$ were selected for this study. The thresholds for accelerations, HiR, and sprint were similar to those reported in previous studies [12, 28]. In this study, the player load is calculated as a downscaled (by a factor of 800) value of the sum of the squared, high pass-filtered accelerometer values for the respective axes (X, Y, and Z): $(X^2 + Y^2 + Z^2)$ / 800 [12]. Test-retest reliability of the sport tracking system is reported earlier, indicating good reliability [12, 28].

**Evaluation of 5-minute periods throughout match time.** To construct an analysis capturing the immediate, dynamic nature of a match for all players, mean values were calculated over consecutive (i.e., moving) 5-min periods for player load and player load per meter, as well as time-motion variables (total distance, accelerations, decelerations, sprint distance, HiR

distance) for comparison, beginning with the first five minutes of the match [25, 29]. The second 5-min period lasts from the second to the sixth minute, and so on. This method is argued to provide a more accurate representation of the distances covered by players [29]. These 5-min periods were used to investigate patterns of player load and locomotor variables throughout match time. The similarities of patterns were then quantified between positions and patterns were evaluated across variables.

## Statistical analysis

All data processing and statistical analysis was performed in Matlab R2019b version 9.7.0.1190202 (Mathworks, Natick, MA, USA). For each match, data for each variable (player load, player load per meter, total distance, accelerations, decelerations, sprint distance, HiR distance) were averaged within positions if there was data from multiple players at the same position, yielding a single time series per variable for each position measured in that match. These data were then converted to z-scores, to facilitate the direct comparison of patterns, disregarding absolute magnitudes. Finally, the z-scores were averaged across all matches for each position, resulting in one time series for each position for each variable. The degree of similarity of patterns between positions was assessed with cross-correlations. For statistical purposes, the break in the time series caused by halftime was disregarded (i.e., the data were treated as continuous for the duration of playing time). Linearity was assessed visually using scatter plots. Cross-correlations were calculated for every position pair for $n$-1 lags at either side of zero, where $n$ = 82, the number of moving 5-min windows in a 90-min match (41 5-min windows in each 45-min half). To best represent the development of player load and time-motion variables across positions throughout match time, the correlation at zero lag (with 95% confidence interval and p-value) is presented. For comparison, maximum correlations and corresponding lags are also reported. A negative lag means that the first time series (player position in table columns) shifts to the left relative to the second time series (player position in table rows). The level of statistical significance was set at $\alpha$ = .05. Correlation values were interpreted categorically as trivial (0–0.1), low (0.1–0.3), moderate (0.3–0.5), high (0.5–0.7), very high (0.7–0.9), or nearly perfect (0.9–1) using the scale presented by Hopkins et al. [30].

## Results

Overall, we observed a distinct pattern in player load throughout match time (Fig 1, black line). The pattern shows peaks at seemingly regular intervals (~15 min), each followed by a period of lower load, typically declining until the next peak. This pattern was clear in all positions (Fig 1A, colored lines), and could also generally be observed in the majority of individual matches (Fig 2). The cross-correlation analysis (Table 1) supports the visual evidence, indicating very high to nearly perfect correlations (range 0.88–0.95, all p < .001) in all position pairs, all having the highest correlation at zero lag. The same pattern was evident for player load per meter, both overall (Fig 1B, black line) and in all positions (Fig 1B, colored lines), with nearly perfect correlation values (range 0.93–0.97, all p < .001; Table 1) in all position pairs, all having the highest correlation at zero lag.

For total distance, no distinct pattern throughout match time was evident (Fig 3A, black line). However, the patterns for all positions appear to follow each other reasonably well (Fig 3A, colored lines), which is reflected in high to very high correlation values (range 0.65–0.89, all p < .001; Table 1), with all position pairs again having the highest correlation at zero lag.

For accelerations, no specific pattern was evident throughout match time (Fig 3B, black line), but the different positions appear to follow roughly similar patterns (Fig 3B, colored lines). Further, correlation values were moderate to high (range 0.32–0.64, all p ≤ .005; S1

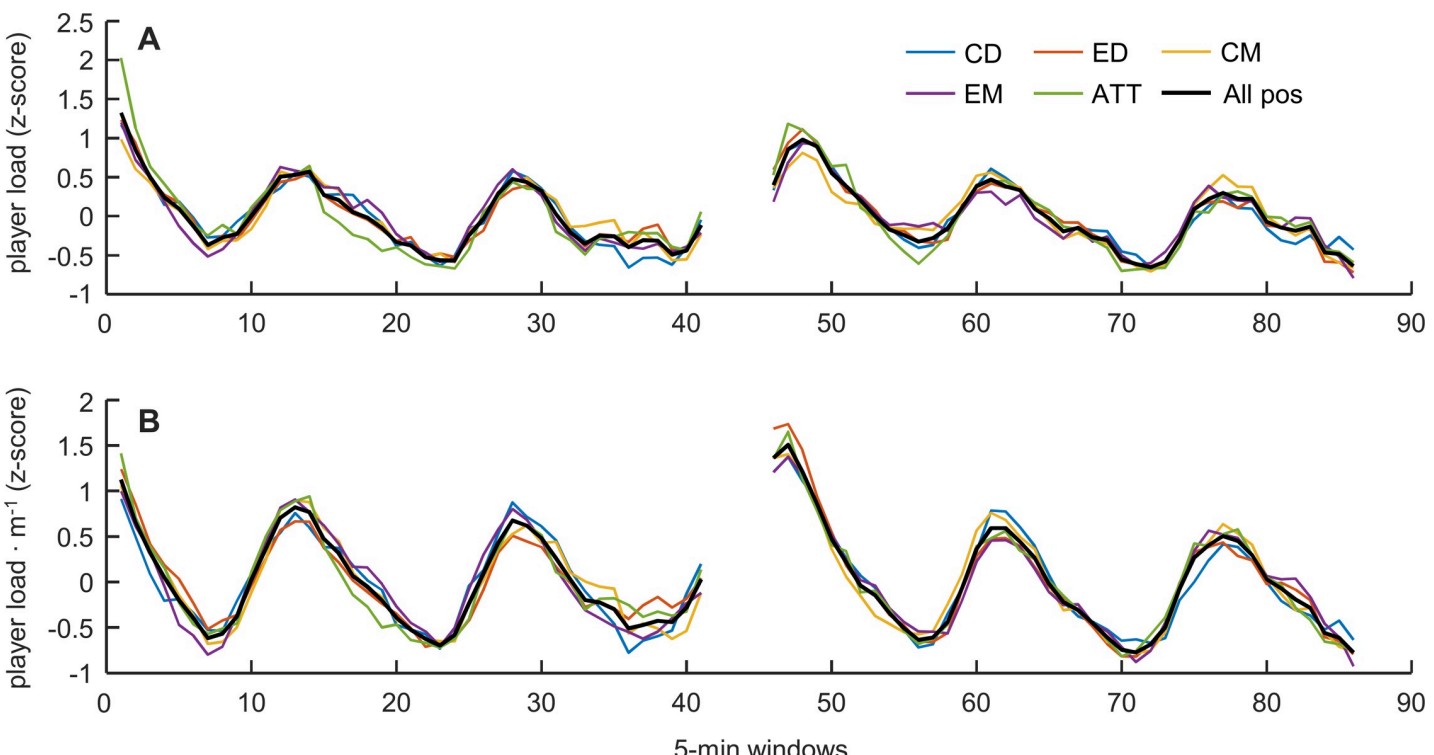

**Fig 1. Mean values (z-scores) of player load and player load per meter in 5-min moving windows throughout match time across all matches (*n* = 34) for each position (colored lines) and for all positions combined (black line).** A: player load; B: player load per meter.

Table), with all but one position pair having the highest correlation at zero lag (EM vs. CD highest absolute correlation 0.56, lag -27; S2 Table). For decelerations, again no specific pattern was evident throughout match time (Fig 3C, black line), but the different positions sporadically follow roughly similar patterns (Fig 3C, colored lines). Correlation values were low to high (range 0.18–0.65, all but one p ≤ .005; S1 Table), with more than half of all position pairs having the highest correlation at zero lag (highest correlation absolute range 0.32–0.65, lag -4–44; S2 Table).

For sprint distance and HiR distance, no specific pattern throughout match time could be discerned in either variable (Fig 3D and 3E, black lines). Further, the patterns for the different positions do not follow each other well (Fig 3 and 3E, colored lines). In line with this, trivial to moderate correlation values were found for sprint distance (absolute range 0.02–0.38, p < .05 in three position pairs, two having the highest correlation at zero lag; highest correlation absolute range 0.29–0.53, lag -29–54 [S1 and S2 Tables]), whereas trivial to moderate (one high) correlation values were found for HiR distance (absolute range 0.01–0.52, p < .05 in three position pairs, two having the highest correlation at zero lag; highest correlation absolute range 0.32–0.52, lag -28–32 [S1 and S2 Tables]).

## Discussion

The primary aim of this study was to explore the patterns of player load with analyses from moving 5-min windows in an elite soccer team. Further, the secondary aim was to compare the player load patterns between different positions within the same team. The main finding was the distinct player load pattern with three "high-load periods" in each half, separated by "lower-load periods". The player load patterns were relatively similar between positions and

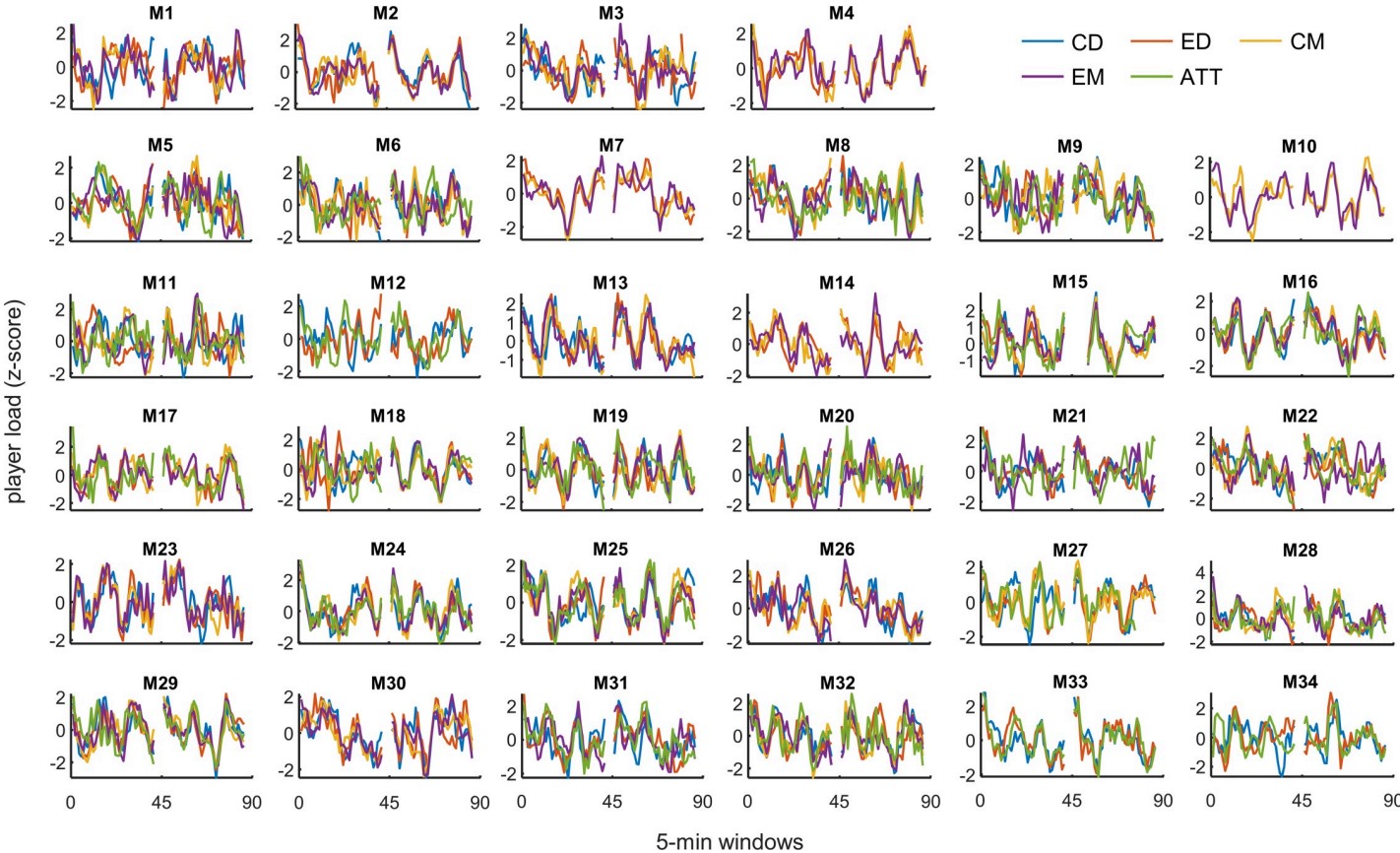

**Fig 2. Mean player load (z-scores) in 5-min moving windows throughout match time for all measured positions per match.** M: match number.

occurred at approximately the same time points during the majority of matches. These novel findings will be discussed with two points of departure: the team's pacing strategy from a physical viewpoint and from a perspective based on interpersonal coordination between player positions.

## Player load patterns and pacing

The use of 5-min moving averages to analyze within-match player load patterns in this study allowed us to study the players' "pacing strategies" (i.e., distribution of player load and related locomotor activities) in more detail than in previous studies evaluating simulated soccer matches [27] and English championship matches [16] by dissection into 15-min periods. The present results show distinct player load patterns with three "high-load periods" in each half of the match (Fig 1), separated by "lower-load periods", in most of the matches (Fig 2), which differs from patterns found in research on English championship players [16]. Although the new methodology for analyzing player load used in the present study provides novel information about high- and lower-load periods of the soccer matches, these distinct patterns found in almost all matches were rather surprising since differences between the opponents' level and tactics should rationally have influenced player load patterns between matches. In addition, the player load would also largely be determined by the players' decision-making about opportunities to become engaged in play. One likely explanation of this apparent player load pattern is that this study investigated one of the top-ranked clubs in the Norwegian top division at

**Table 1. Cross-correlations [95% CI] of mean position values (z-scores) across all matches (*n* = 34) at zero lag for player load, player load per meter, and total distance.**

| | CD | ED | CM | EM | ATT |
|---|---|---|---|---|---|
| *Player load* | | | | | |
| CD | --- | | | | |
| ED | 0.95 [0.92, 0.96] | --- | | | |
| CM | 0.93 [0.89, 0.95] | 0.94 [0.91, 0.96] | --- | | |
| EM | 0.93 [0.89, 0.95] | 0.94 [0.91, 0.96] | 0.93 [0.90, 0.96] | --- | |
| ATT | 0.91 [0.87, 0.94] | 0.95 [0.93, 0.97] | 0.88 [0.83, 0.92] | 0.89 [0.84, 0.93] | --- |
| *Player load per meter* | | | | | |
| CD | --- | | | | |
| ED | 0.93 [0.89, 0.95] | --- | | | |
| CM | 0.95 [0.93, 0.97] | 0.95 [0.92, 0.97] | --- | | |
| EM | 0.95 [0.92, 0.97] | 0.94 [0.91, 0.96] | 0.95 [0.92, 0.97] | --- | |
| ATT | 0.93 [0.90, 0.96] | 0.97 [0.96, 0.98] | 0.96 [0.94, 0.97] | 0.95 [0.92, 0.97] | --- |
| *Total distance* | | | | | |
| CD | --- | | | | |
| ED | 0.88 [0.81, 0.92] | --- | | | |
| CM | 0.80 [0.71, 0.87] | 0.84 [0.76, 0.89] | --- | | |
| EM | 0.89 [0.84, 0.93] | 0.87 [0.81, 0.92] | 0.86 [0.79, 0.91] | --- | |
| ATT | 0.76 [0.65, 0.84] | 0.71 [0.59, 0.81] | 0.65 [0.51, 0.76] | 0.72 [0.59, 0.81] | --- |

CD = central defender; ED = external defender; CM = central midfielder; EM = external midfielder; ATT = attacker. All correlations p < .001. For all correlations, the maximum value occurred at zero lag.

their home arena, where they had the opportunity to "control the match" in most of the matches. Thus, it seems reasonable to ask whether these similar positional fluctuations in player load are typical for this team at their home arena matches where they normally were the dominant team. Therefore, an interesting approach for future studies would be to investigate these patterns with the same moving average-method in teams at different performance levels (i.e., if the investigated team or the opposition controls the match or in teams with different overall tactical dispositions).

Since locomotor actions in soccer are not performed in isolation, consideration of player load as a proxy for "overall external load" might be useful. A previous investigation of player load found high to very high associations between player load and measures of internal training load (TRIMP and sRPE) [18], with internal load being especially related to the volume of accelerations. Barrett et al. [18] found nearly perfect within-subject correlation between player load and heart rate/$VO_2$, but trivial to moderate association for the between-subject correlation on the same variable [19]. Overall, this suggests that the fluctuations in player load found in the present study are also associated with fluctuations in internal load, thereby indicating a physical "pacing pattern" (pattern in distribution of load) employed by the investigated team (Fig 1). These "pacing patterns" were relatively similar between positions and occurred at the same time point during the matches (Fig 2), even though the different positions have different roles during attacks and defense; one single attack gives higher intensities on attacking players, but not for the defending players, and vice versa. However, the time scale with 5-min moving averages is too long to differentiate between high-intensity periods based on one single attack or one defensive stand and normally contain several attacking and defensive actions. Moreover, player load patterns based on moving 5-min windows will give more information about

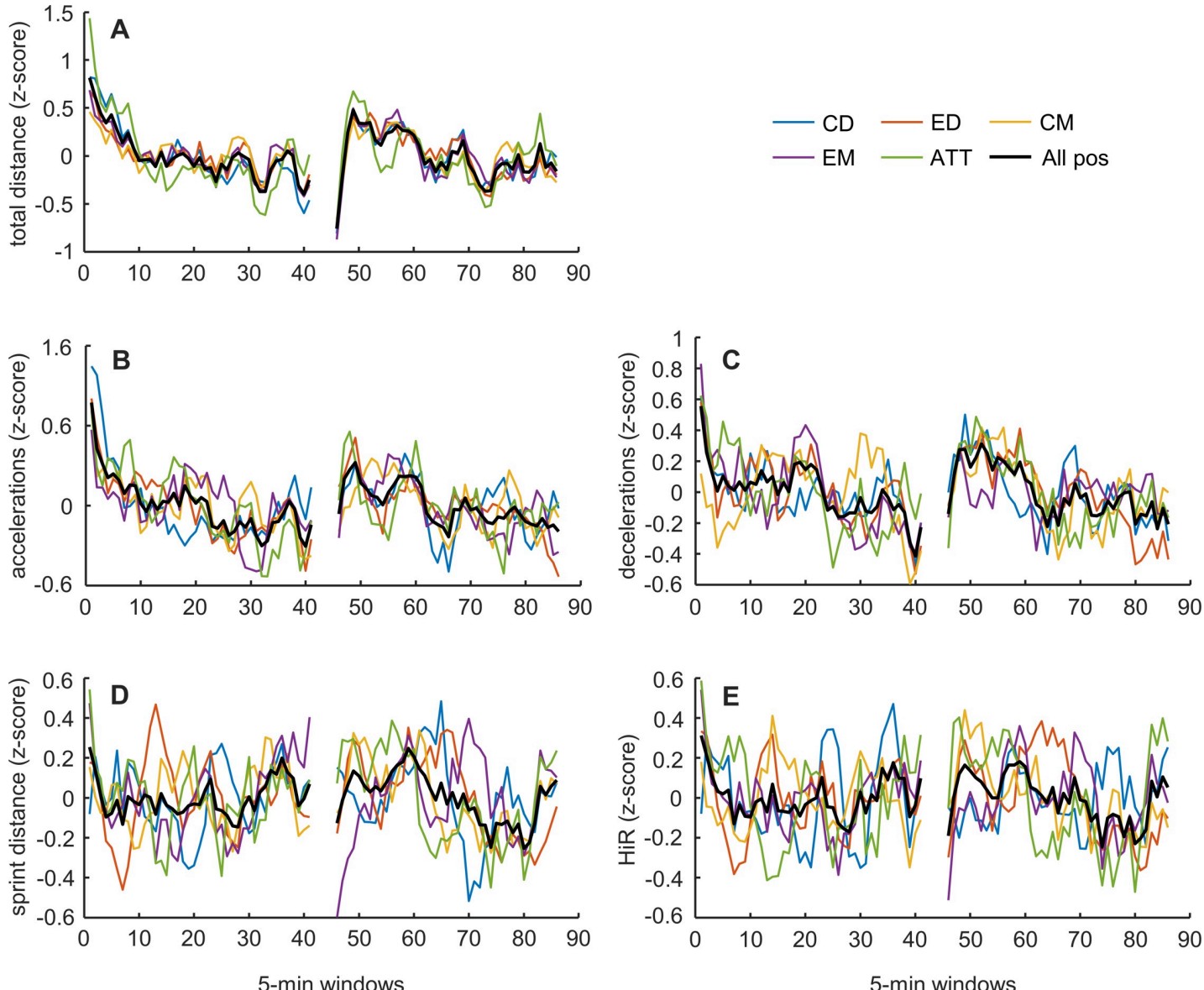

**Fig 3. Mean values (z-scores) of time-motion variables in 5-min moving windows throughout match time across all matches (*n* = 34) for each position (colored lines) and for all positions combined (black lines).** A: total distance; B: accelerations; C: decelerations; D: sprint distance; E: high-intensity running distance (HiR).

the overall load of the match, instead of detailed information about when the team is attacking (more load on offensive players) or defending (more load on defensive players).

The present study shows very high to nearly perfect associations between positional patterns of player load and player load per meter (Table 1). Hence, the periods of high player load are associated with movements on the field that increases the player load per meter, which is shown to be associated with unorthodox movements such as jumping, tackling, collisions, passing, accelerations, decelerations etc., movement which are common for soccer and detected when triaxial accelerometers are employed [12, 13]. Although this study found differences in the absolute values of the highest and lowest player load periods in the presented results, there were no positional differences in the pattern of increase and decrease of player load throughout the matches (Fig 1). Thus, the present study is the first to report similarity

across playing positions in the player load patterns throughout matches in male elite soccer players. The use of this approach and the findings from this study may contribute to new hypotheses concerning the patterns of player load and intensity throughout a soccer match. Therefore, before one can conceptualize more in-field applications, different aspects of player load patterns should be investigated further.

Whereas other investigations show a considerable heterogeneity in the within-match pattern of total distance, HiR and sprint across studies [9, 20, 21], in this study, total distance was the variable besides player load and player load per meter which displayed the highest correlation between positions, with no lag between positional patterns (Table 1). Regardless of this, the patterns of total distance for the different positions do not follow the same distinct pattern as the player load variables. For accelerations and deceleration, no specific pattern was evident throughout match time (Fig 3B and 3C), but the different positions appear to follow roughly similar patterns with correlations ranging from low to high (S1 Table). For sprint and HiR distance, the present study shows no meaningful similarities between positions, with negligible to moderate cross-correlations (S1 Table). These findings are similar to those from other studies investigating high-intensity patterns [6, 7]. In the present results, the patterns of the different HiR and sprint distance throughout match time show heterogeneity; patterns of sprint and HiR distance show that high-intensity periods occur at different times both between matches and between positions. These differences could be caused by different tactical elements, opponents playing style, pacing strategies, score line, and team formation, which would all affect the players' ability to regulate their physical effort and maintain work rates at appropriate levels [22].

## Player load and interpersonal coordination patterns

The observed in-phase pattern for player load in this study is also interesting from perspectives of interpersonal coordination patterns, and it demonstrates that the interaction in player load between the team's subunits probably is more complex than the behavior of each individual player considered separately [31, 32]. Specifically related to soccer, the actions of one player or a player subunit (e.g., attackers, midfielders, defenders) cause re-actions and adjustments from other players or player subunits to stabilize performance, and these adjustments interact and influence player load collectively. The emergence of the synchronized player load patterns between subunits is likely self-organized to improve team performance and is a result of the interactions of a player's constraints and information exchange within their own team and those imposed by the opponent. What type of constraints and information that evolves in spontaneous self-organization and synchronization of player load is not easy to identify, but might be easily understood intuitively. Examples of such constraints in soccer could be other players' positions and movements, position and speed of the ball, tactical decisions, fatigue, etc. According to the rationale by Haken and Portugali [33], if the meaning of a player's action is understood (information exchange), it triggers action and changes the structure or behavior (player load) in the whole team. E.g., the reaction of players on the action of another depends on the success or failure (information) of that action. The interesting finding of the present study is that, even though each action's success or failure may occur randomly, the player load pattern that evolves seems very stable. Thus, the interpersonal patterns of coordination of player load in a soccer team might be modelled as an open complex dynamical system at a behavioral level of analysis, as suggested in evolutionary game theory [34]. Given the stable player load pattern over various matches, even though a soccer match is the complex combination of actions by individuals, no individual player (or subunit) seems to initiate or control the behavior of the match. In other words, each player is enslaved in a self-organized system that

at the same time consists of all these same players. This self-organized system could be affected by the fact that this study investigated one of the top-ranked clubs at their home arena, which could have produced a more consistent player load pattern due to typically being the dominant team.

## Limitations

Since this study investigated one of the top-ranked clubs at their home arena, it is possible a more consistent player load pattern was produced due to typically being the dominant team. It is unclear to what extent the results will replicate across teams or if they are particular to either the investigated team or e.g., teams sharing certain characteristics. This study did not investigate differences between various tactical elements, opponents' playing styles, ball in versus out of play, score line, or team formations, which could all affect the players' ability to regulate their physical effort and maintain work rate profiles. Differences in measurement technology makes it difficult to compare player load variable between different tracking systems (or even different versions of the same system), since differences in measurement technology could partly account for eventual discrepancies between the values registered in this study and other studies. Hence caution is required when comparing analyses of football match activities across studies.

## Conclusion

This study demonstrated similarity in player load patterns between positions in elite soccer matches. The novelty is the clear pattern which consists of three high-load periods in both halves, where these "high load" periods are followed by periods with reduced load. The present study did not find similar unambiguous patterns on any of the locomotor variables. The evident pattern in player load indicates a physical "pacing pattern" employed by the team. These "pacing patterns" were relatively similar between positions and occurred at the same time points during the matches over three successive seasons. From the perspective of interpersonal coordination patterns, these synchronized player load patterns between positions are likely self-organized to improve team performance and are a result of the interactions of the players' constraints and information exchange within their own team and those imposed by the opponents. It should be noted that a more consistent player load pattern might have been produced due to the investigated team being a top-ranked club playing home matches.

## Practical applications

Since this study is the first to report this distinct pattern of player load it is important that more studies of player load patterns are conducted, in teams at different performance levels before in-field applications can be firmly conceptualized. Considering the previously reported high association between player load and internal training load, it could be argued that coaches might want to regulate player load in training for an overreaching effect. This could eventually allow for a more aggressive pacing strategy, shortening the lower-load periods and hence putting more pressure on the opposition. However, an approach like this must be cautious against overloading. During matches, coaches can also use the method proposed here in real-time to monitor if certain players or position groups appear to be "out of sync" with the rest of the team.

## Supporting information

**S1 Table. Cross-correlations [95% CI] of mean position values (z-scores) across all matches (*n* = 34) at zero lag for accelerations, decelerations, sprint distance, and high-intensity**

**running distance.** CD: central defender; ED: external defender; CM: central midfielder; EM: external midfielder; ATT: attacker. For p-values, bold text indicates significance at α = .05. (DOCX)

**S2 Table. Maximum cross-correlations (corresponding lag) of mean position values (z-scores) across all matches (*n* = 34) for accelerations, decelerations, sprint distance, and high-intensity running distance.** CD: central defender; ED: external defender; CM: central midfielder; EM: external midfielder; ATT: attacker. A negative lag means that the first time series (player position in table columns) shifts to the left relative to the second time series (player position in table rows).
(DOCX)

**S1 Dataset.**
(XLSX)

## Acknowledgments

We thank the players for their efforts throughout the period.

## Author Contributions

**Conceptualization:** Terje Dalen, Tore Kristian Aune, Geir Håvard Hjelde, Øyvind Sandbakk, David McGhie.

**Data curation:** Terje Dalen, David McGhie.

**Formal analysis:** Terje Dalen, Øyvind Sandbakk, David McGhie.

**Investigation:** Terje Dalen, Geir Håvard Hjelde, David McGhie.

**Methodology:** Terje Dalen, Øyvind Sandbakk, David McGhie.

**Project administration:** Terje Dalen, David McGhie.

**Resources:** Terje Dalen.

**Supervision:** Terje Dalen.

**Validation:** Terje Dalen.

**Visualization:** Terje Dalen, David McGhie.

**Writing – original draft:** Terje Dalen, Tore Kristian Aune, Øyvind Sandbakk, David McGhie.

**Writing – review & editing:** Terje Dalen, Tore Kristian Aune, Gertjan Ettema, Øyvind Sandbakk, David McGhie.

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
