## [Decision Letter · Decision Letter 0]

23 Jun 2020

PONE-D-20-16658

Player load in male elite soccer matches: comparisons of patterns between positions and matches

PLOS ONE

Dear Dr. Dalen,

Thank you for submitting your manuscript to PLOS ONE. After careful consideration, we feel that it has merit but does not fully meet PLOS ONE’s publication criteria as it currently stands. Therefore, we invite you to submit a revised version of the manuscript that addresses the points raised during the review process.

The novelty of this study is incontestable, but adding deceleration evaluation in the data reported, as suggested by one reviewer, could increase the relevance of the manuscript. Beyond the minor comments raised during the review process, some practical considerations are also to be added.

We look forward to receiving your revised manuscript.

Kind regards,

Laurent Mourot

Academic Editor

PLOS ONE

Journal Requirements:

'The authors have declared that no competing interests exist.'

We note that one or more of the authors are employed by a commercial company: Rosenborg FC.

Reviewers' comments:

Reviewer's Responses to Questions

**Comments to the Author**

1. Is the manuscript technically sound, and do the data support the conclusions?

Reviewer #1: Yes

Reviewer #2: Yes

2. Has the statistical analysis been performed appropriately and rigorously? 

Reviewer #1: Yes

Reviewer #2: Yes

3. Have the authors made all data underlying the findings in their manuscript fully available?

Reviewer #1: Yes

Reviewer #2: Yes

4. Is the manuscript presented in an intelligible fashion and written in standard English?

Reviewer #1: Yes

Reviewer #2: Yes

5. Review Comments to the Author

Reviewer #1: The manuscript is overall well-written and investigates an interesting and novel aspect in soccer. I have just some minor suggestions to improve the manuscript. These are listed below.

Abstract

• Lines 28-31: please rephrase, not clear.

• Please populate the results section

• “player load pattern” is maybe not clear in the abstract.

Introduction

• The introduction is overall good and provide a sufficient rationale for the study. However, I’d suggest shortening it and being more straightforward while introducing why this study is needed. Such a quite long introduction led to the dependent parameters to be dispersed. Please clearly address them.

Methods

• The dependent parameters have not been clearly defined. What was the speed-zone for each?

• Statistical analysis: the previous information (introduction and methods) does not lead me to figure out the statistical approach the Authors used. For example, at some time I would have expected a repeated-measure ANOVA, while the Authors used (appropriately!) cross-correlation. Please be clearer in the previous sections, so I could clearly understand why this approach is suitable.

Discussion

• High-intensity or high-accumulated load? Please consider rewording it.

• I’d suggest, at the beginning of each paragraph, providing a clear interpretation of the results, so that a reader could easily understand what really happened.

• How ball possession and ball in vs out of play may have influenced the results?

• I believe a limitations section is needed.

• Please provide possible practical applications of these results.

Reviewer #2: Having read the manuscript "Player load in male elite soccer matches: comparisons of patterns between positions and matches", I recognize an interesting contribution for the current state of the art in this specific topic, however some major topics should be reviewed or added. The text shows clarity and flow but there is a lack of proper practical applications of the current paper. Hence, I have recommended major revision to improve further text clarity before I can consider recommending it for publishing, according the following comments:

First, I recommend that you change your title. In fact, you highlight the study of player load, but you also analyse other external load variables, such as total distance, high-speed running distance and acceleration. Maybe use external load instead of player load or to identify all variables analysed.

A second comment or question that I want to make is why do you not use deceleration? Because there are recent studies that highlight the importance of this variables along with acceleration. Is it possible for you to add some information regarding deceleration? I think that would be very interesting and useful for coaches, staff, or researchers.

Introduction

L42 - which association do you want to refer?

L47 - and training as well.

L59 - I suggest a different approach regarding the use of the word strain. There are other concepts in soccer analysis that include strain, known as training strain or training strain index, or training strain workload.

L61 - In the first line of the introduction, you start to mention soccer. In my opinion, there is no need to cite a study that concerns a different sport such basketball.

L66-67 - I suggest changing this sentence for: "To date, monitoring external training and match load measures in soccer has tended to rely on results based on locomotor activities". You must check English.

L69 - What studies? You must cite them.

L69 - I suggest putting this abbreviation after high-intensity running.

L70 - In the beginning of the sentence you refer "some studies" but then you only mention one (reference 9). You must review this.

L76-77 – “A less researched component of soccer matches is the players’ number of accelerations“ - I do not agree with this statement. Although I know your paper Terje Dalen, Håvard Lorås, Geir Håvard Hjelde, Terje Naess Kjøsnes & Ulrik Wisløff (2019): Accelerations – a new approach to quantify physical performance decline in male elite soccer?, European Journal of Sport Science, DOI: 10.1080/17461391.2019.1566403 where you mention that, there are a lot of studies regarding acceleration topic. You probably know the following paper regarding acceleration and deceleration: Harper, D.J., Carling, C. & Kiely, J. High-Intensity Acceleration and Deceleration Demands in Elite Team Sports Competitive Match Play: A Systematic Review and Meta-Analysis of Observational Studies. Sports Med 49, 1923–1947 (2019). https://doi.org/10.1007/s40279-019-01170-1

You may want to say that there still is a need to better study this variable in order to provide practical applications or insight for soccer science and coaches.

Methods – Subjects – I suggest to use Participants instead of subjects.

What were the inclusion criteria of the participating players? How many matches did the players participate over the 3 seasons? How many minutes? Did you control these variables? This information should be added.

L125 – Instead of using “Some players played”, I suggest changing for "Some players participated" to avoid word repetition.

Study design and methodology

L138 – When you mention “high-speed running, you may use the abbreviation.

L215 - What was the result that you mention? This kind of sentence usually fix better for discussion section. In the results section, you must be clear, concise and objective.

L217 - what do you mean with the positions appear to follow each other only very roughly? It I not clear.

L231 – Discussion - You should start your discussion with the aim of your study and then by presenting the main results.

L240-242 - Can you provide any reference to support your statement? Because what you are saying is a huge statement for all studies that analysed external load variables without including player load.

L243-244 - I suggest that you add some information about what does mean "pacing strategies" in this study. This is not clear even in the introduction section.

L247 - …”which differs from patterns found in other research [17,28].” I suggest that you provide more knowledge regarding the teams analysed in those studies [17,28], the duration of the season analysed and other contextual variables that could help the reader to understand the differences between studies.

L269 – “Pacing pattern” - This designation needs to be clarified in the text.

L284 – “…common for team sport athletes” - Your study is regarding soccer players. Therefore, you should mention specific movements in soccer and not generalize them for sports.

L290 - …”patterns of player load and intensity throughout a soccer match.” Until this point, you still does not provide what was the patterns that you found in discussion section.

L295 – “Locomotor activities” is a term too vague because it can include many different activities at low and high intensity. I suggest that you clear specifically what you mean regarding the cited studies [9,21,22].

L301-304 - Is this sentence referring to the studies 6 and 7? It is not clear.

Conclusions

What are the practical applications for soccer coaches, staff members, or soccer science?

For example, can you provide some recommendations for soccer training? Also, you must point some limitations of this study.

6. PLOS authors have the option to publish the peer review history of their article (what does this mean?). If published, this will include your full peer review and any attached files.

Reviewer #1: Yes: Giuseppe Coratella

Reviewer #2: Yes: Rafael Franco Soares Oliveira

---

## [Author Response · Author response to Decision Letter 0]

10 Aug 2020

PONE-D-20-16658

Player load in male elite soccer matches: comparisons of patterns between positions and matches

PLOS ONE

Academic Editor comments to the author:

Thank you for submitting your manuscript to PLOS ONE. After careful consideration, we feel that it has merit but does not fully meet PLOS ONE’s publication criteria as it currently stands. Therefore, we invite you to submit a revised version of the manuscript that addresses the points raised during the review process.

The novelty of this study is incontestable, but adding deceleration evaluation in the data reported, as suggested by one reviewer, could increase the relevance of the manuscript. Beyond the minor comments raised during the review process, some practical considerations are also to be added.

Response to Academic Editor: Thank you for letting us revise our manuscript. We have now included deceleration evaluation in the manuscript and included some practical consideration as requested by you and the reviewers. We thank you and the reviewers for helping us get our paper better and do hope you find the present version of the manuscript acceptable for publication. 

Review Comments to the Author

Reviewer #1: The manuscript is overall well-written and investigates an interesting and novel aspect in soccer. I have just some minor suggestions to improve the manuscript. These are listed below.

Abstract

# Comment 1: Lines 28-31: please rephrase, not clear. 

# Response 1: We apologize for the unclear phrasing. We have now attempted to simplify and clarify this information in the abstract to make it easier to understand the essence of what was done (within the word limits of the abstract)

In the abstract, the sentence now reads “Data for each variable (player load, player load per meter, total distance, accelerations, decelerations, sprint distance, high-intensity running distance) were averaged within positions in each match, converted to z-scores and averaged across all matches, yielding one time series for each variable for each position.”

We have also added a minor specification in the description in the methods.

# Comment 2: Please populate the results section 

# Response 2: We apologize for the scarceness of data presented in the abstract. We have now added data from the remaining variables in the results section for comparison. Due to these additions, as well as the changes made from comment 1, we have made a number of minor language/syntax edits throughout the abstract to keep the length within the 300-word limit.

# Comment 3: “player load pattern” is maybe not clear in the abstract.

# Response 3: We apologize for not defining this properly from the beginning. We have now amended the abstract to clarify this for the reader already in the first sentence: “Our primary aim was to explore the development of player load throughout match time (i.e., the pattern) using moving 5-min windows in an elite soccer team and our secondary aim to compare player load patterns between different positions within the same team.”

# Comment 4: The introduction is overall good and provide a sufficient rationale for the study. However, I’d suggest shortening it and being more straightforward while introducing why this study is needed. Such a quite long introduction led to the dependent parameters to be dispersed. Please clearly address them.

# Response 4: We have now shortened the text where we felt it was appropriate and moved some information between paragraphs in order to present information more straightforwardly, while still keeping the buildup of information intact. With the exception of player load, the dependent parameters are all presented in the third paragraph. We have now also specified the main variables and the supporting variables in the last paragraph of the introduction, when presenting our aims.

If the reviewer insists, we are of course willing to make further efforts to shorten the text and/or clarify the dependent parameters.

Method

# Comment 5: The dependent parameters have not been clearly defined. What was the speed-zone for each?

# Response 5: We apologize for this oversight and have now completed the sentence with speed zones to also include the sprint category. The text now reads: “Accelerations and decelerations were recorded when they reached limits of 2 m.s-2 and -2 m.s-2, respectively, and a high-speed running category of >19.8 km.h-1 and sprint category of >25.2 km.h-1 were selected for this study. The thresholds for accelerations, HiR, and sprint were similar to those reported in previous studies [12,29].”

# Comment 6: Statistical analysis: the previous information (introduction and methods) does not lead me to figure out the statistical approach the Authors used. For example, at some time I would have expected a repeated-measure ANOVA, while the Authors used (appropriately!) cross-correlation. Please be clearer in the previous sections, so I could clearly understand why this approach is suitable.

# Response 6: We appreciate you recognizing the appropriateness of our statistical approach, and also understand that the reader could benefit from some “foreshadowing” of what is coming. We have now added some specifications in the introduction and the methods to make the reasoning for the statistical approach clearer for the reader.

At the first mention of patterns, we define our use of patterns, consistent with the change made to the abstract: “In previous analyses of soccer matches, considerable heterogeneity has been observed in the within-match development of locomotor activities throughout match time (i.e., the pattern) across studies [6,9, 20-26]”

In the second to last paragraph, we further specify the need to quantify pattern similarities throughout the duration of matches: “More instantaneous analyses of player load and the corresponding activities during soccer matches would logically show a variable “pacing” influenced by e.g., tactical elements, player position, and the level of the opponents. In order to quantify this across e.g., positions, the similarity of patterns throughout the duration of matches must be analyzed. Long term analyses of such data…”

In the methods, before the statistical analysis, we again specify the quantification of pattern comparisons: “These 5-min periods were used to investigate patterns of player load and locomotor variables throughout match time. The similarities of patterns were then quantified between positions and patterns were evaluated across variables.”

Discussion

# Comment 7: High-intensity or high-accumulated load? Please consider rewording it.

# Response 7: We see that “high-intensity” might cause confusion. However, what we present is not accumulated load, so we do not feel that is a suitable phrasing either. Since our method was based on analyzing 5-min periods, we have kept “periods”, but have now changed “high-intensity periods” and “low-intensity periods” to “high-load periods” and “lower-load periods” throughout the manuscript.

# Comment 8: I’d suggest, at the beginning of each paragraph, providing a clear interpretation of the results, so that a reader could easily understand what really happened.

# Response 8: Thank you for this suggestion. We agree that this is a good way of introducing the discussion paragraphs and have made changes in an effort to make the topics of the discussion clearer to the reader. However, due to the content of the different discussion paragraphs, it is not always direct interpretations of results, but also methodological explanations/arguments. 

# Comment 9: How ball possession and ball in vs out of play may have influenced the results?

# Response 9: This is difficult to answer specifically. In both theory and practice, many factors could have influenced the results. However, the time scale with 5-min moving averages is likely too long to differentiate between ball possession and ball in vs. out of play. Nevertheless, ball in vs. out of play is now mentioned in the new limitations section.

With that said, we feel that the fact that the player load that occurs is so clear, similar between positions, and consistent in when it occurs, despite all the factors that might influence the results, is a testament to the strength of our main findings.

# Comment 10: I believe a limitations section is needed.

# Response 10: We had initially addressed some limitations in the discussion but did not present them collectively. A limitation section is now included.

# Comment 11: Please provide possible practical applications of these results.

# Response 11: Since this is the first study to report this kind of pattern, it is a bit difficult to recommend practical applications based on the finding. However, we agree that it is an important part of research, and have now provided possible practical applications, to the extent that we are able, in a section at the end.

Reviewer #2: Having read the manuscript "Player load in male elite soccer matches: comparisons of patterns between positions and matches", I recognize an interesting contribution for the current state of the art in this specific topic, however some major topics should be reviewed or added. The text shows clarity and flow but there is a lack of proper practical applications of the current paper. Hence, I have recommended major revision to improve further text clarity before I can consider recommending it for publishing, according the following comments:

# Comment 1: First, I recommend that you change your title. In fact, you highlight the study of player load, but you also analyse other external load variables, such as total distance, high-speed running distance and acceleration. Maybe use external load instead of player load or to identify all variables analysed.

# Response 1: We understand this recommendation but would prefer to keep player load as the focus of the title, since it is both the main aim and the main finding of the study and the remaining variables are primarily supporting variables for the sake of comparison. However, we have made some minor changes to the title, which is now: “Player load in male elite soccer: comparisons of patterns between matches and positions”. If the reviewer insists, we are of course willing to take another look at this.

# Comment 2: A second comment or question that I want to make is why do you not use deceleration? Because there are recent studies that highlight the importance of this variables along with acceleration. Is it possible for you to add some information regarding deceleration? I think that would be very interesting and useful for coaches, staff, or researchers.

# Response 2: Thank you for this input. We initially decided to omit decelerations due to the number of variables analyzed, prioritizing the most commonly investigated variables for comparisons. We have now added deceleration info and data to the methods section, results section, figure 3 (new panel C; old panel C becomes D, and D becomes E), supporting tables S1 and S2, and the S3 dataset. The results of the deceleration analysis essentially did not change our outcome, as it was very much in line with the results from accelerations. Nevertheless, the data is now there for coaches, staff, or researchers to evaluate.

Introduction

# Comment 3: L42 - which association do you want to refer?

# Response 3: Here we are using the noun “association football”, more commonly known as soccer or football.

# Comment 4: L47 - and training as well.

# Response 4: Thank you for pointing out this oversight. We have now changed this to “training and matches”.

# Comment 5: L59 - I suggest a different approach regarding the use of the word strain. There are other concepts in soccer analysis that include strain, known as training strain or training strain index, or training strain workload.

# Response 5: We recognize that the use of the word “strain” here can cause unnecessary confusion with established concepts. We have now changed “physical strain” to “physical exertion”.

# Comment 6: L61 - In the first line of the introduction, you start to mention soccer. In my opinion, there is no need to cite a study that concerns a different sport such basketball.

# Response 6: We agree, the studies from Australian football and basketball are now deleted.

# Comment 7: L66-67 - I suggest changing this sentence for: "To date, monitoring external training and match load measures in soccer has tended to rely on results based on locomotor activities". You must check English.

# Response 7: We agree with this suggestion and have changed the sentence accordingly.

# Comment 8: L69 - What studies? You must cite them.

# Response 8: We apologize for not providing the references. These are now added.

# Comment 9: L69 - I suggest putting this abbreviation after high-intensity running.

# Response 9: We agree and have now changed this according to the comment.

# Comment 10: L70 - In the beginning of the sentence you refer "some studies" but then you only mention one (reference 9). You must review this.

# Response 10: We apologize for this omission. We have now added another reference: “Bradley, P. S., et al. (2009). "High-intensity running in English FA Premier League soccer matches." J Sports Sci 27(2): 159-168.”

# Comment 11: L76-77 – “A less researched component of soccer matches is the players’ number of accelerations“ - I do not agree with this statement. Although I know your paper Terje Dalen, Håvard Lorås, Geir Håvard Hjelde, Terje Naess Kjøsnes & Ulrik Wisløff (2019): Accelerations – a new approach to quantify physical performance decline in male elite soccer?, European Journal of Sport Science, DOI: 10.1080/17461391.2019.1566403 where you mention that, there are a lot of studies regarding acceleration topic. You probably know the following paper regarding acceleration and deceleration: Harper, D.J., Carling, C. & Kiely, J. High-Intensity Acceleration and Deceleration Demands in Elite Team Sports Competitive Match Play: A Systematic Review and Meta-Analysis of Observational Studies. Sports Med 49, 1923–1947 (2019). https://doi.org/10.1007/s40279-019-01170-1

You may want to say that there still is a need to better study this variable in order to provide practical applications or insight for soccer science and coaches.

# Response 11: We apologize for our unclear phrasing, which may have led to a misunderstanding. “Less researched” in this line was in comparison to variables such as HiR and sprints, which are abundant in the literature, and not meant to imply that there is no or very little research done on accelerations/decelerations. We have now changed this sentence to: “A component of soccer matches that has received relatively less attention is the players’ number of accelerations and decelerations [20] …” Reference [20] is Harper et al….

To highlight that we do not mean to imply that no acceleration/deceleration research exists, we have now added a reference to the paper you mention. 

# Comment 12: Methods – Subjects – I suggest to use Participants instead of subjects.

# Response 12: We agree and have changed this according to the comment.

# Comment 13: What were the inclusion criteria of the participating players? How many matches did the players participate over the 3 seasons? How many minutes? Did you control these variables? This information should be added.

# Response 13: We apologize if this was unclear, but would argue that the inclusion criteria are already described in the “participants” paragraph in the methods: data from a player was only included if a player completed the entire match, and goalkeepers were excluded; further, only home matches were included (due to technical measurements reasons: the system was installed at the home arena), and only domestic matches (to exclude a small number of matches against a different level of competition; too few to make meaningful and trustworthy comparisons).

We did not control which players ultimately qualified for the criteria but were rather at the mercy of the coach’s decisions on starting players and substitutions.

One of our aims was to explore positions within a team, and as such, different players in the same position were treated as representing that position in the analysis, since they all played in the same system with the same roles. Since this study is based on a three-season dataset, it is highly variable how many matches each player participated in, but the number of players representing each position and the total number of full matches completed by that group is provided.

If the information provided is still deemed insufficient, we are of course willing to take another look at this.

# Comment 14: L125 – Instead of using “Some players played”, I suggest changing for "Some players participated" to avoid word repetition.

# Response 14: Thank you for pointing this out. We have now changed this according to the comment.

Study design and methodology

# Comment 15: L138 – When you mention “high-speed running, you may use the abbreviation.

# Response 15: This has now been changed according to the comment.

# Comment 16: L215 - What was the result that you mention? This kind of sentence usually fix better for discussion section. In the results section, you must be clear, concise and objective.

# Response 16: Thank you for noting this. We have now removed this type of language from the results section. This sentence now reads “For accelerations, no specific pattern…”.

# Comment 17: L217 - what do you mean with the positions appear to follow each other only very roughly? It I not clear.

# Response 17: We apologize for the unclear phrasing. We have now rephrased this to “…but the different positions appear to follow roughly similar patterns”. We have also tried to make this clear in the other paragraphs in the results.

# Comment 18: L231 – Discussion - You should start your discussion with the aim of your study and then by presenting the main results.

# Response 18: We apologize for this oversight. We now repeat our aims at the start of the discussion.

# Comment 19: L240-242 - Can you provide any reference to support your statement? Because what you are saying is a huge statement for all studies that analysed external load variables without including player load.

# Response 19: Thank you for noting this. We agree and realize that this read as a much stronger statement than we intended. Therefore, we have amended this statement to: “Since locomotor actions in soccer are not performed in isolation, consideration of player load as a proxy for “overall external load” might be useful.”

Also, this sentence has now been moved to the following paragraph.

# Comment 20: L243-244 - I suggest that you add some information about what does mean "pacing strategies" in this study. This is not clear even in the introduction section.

# Response 20: We apologize for the lack of clarity, and have now tried to specify this in the introduction: “More instantaneous analyses of player load and the corresponding activities during soccer matches would logically show a variable “pacing” influenced by e.g., tactical elements, player position, and the level of the opponents. (…) Long term analyses of such data and the relationships to changes in tactics, different opponents, and match outcome have the potential to provide imperative understanding of how the team and the players in different positions distribute the load (i.e., “pacing strategies”) during different types of matches.”

# Comment 21: L247 - …”which differs from patterns found in other research [17,28].” I suggest that you provide more knowledge regarding the teams analysed in those studies [17,28], the duration of the season analysed and other contextual variables that could help the reader to understand the differences between studies.

# Response 21: We apologize for omitting this information. We have now added contextual information to both the sentence you mention and the preceding sentence: “The use of 5-min moving averages to analyze within-match player load patterns in this study allowed us to study the players’ “pacing strategies” (i.e., distribution of player load and related locomotor activities) in more detail than in previous studies evaluating simulated soccer matches [28] and English championship matches [17] by dissection into 15-min periods. The present results show distinct player load patterns with three “high-load periods” in each half of the match, separated by “lower-load periods”, in most of the matches, which differs from patterns found in research on English championship players [17].”

# Comment 22: L269 – “Pacing pattern” - This designation needs to be clarified in the text.

# Response 22: We apologize for the lack of clarity. In line with the answer to comment 20, we have now specified this: “Overall, this suggests that the fluctuations in player load found in the present study are also associated with fluctuations in internal load, thereby indicating a physical “pacing pattern” (pattern in distribution of load) employed by the investigated team”

# Comment 23: L284 – “…common for team sport athletes” - Your study is regarding soccer players. Therefore, you should mention specific movements in soccer and not generalize them for sports.

# Response 23: We agree, changed to: “…movement which are common for soccer…”

# Comment 24: L290 - …”patterns of player load and intensity throughout a soccer match.” Until this point, you still does not provide what was the patterns that you found in discussion section.

# Response 24: Although we respectfully disagree with this statement, we apologize if we did not make our descriptions clear enough. The pattern is described as the main finding in the very first paragraph in the discussion (“The main finding was the distinct player load pattern with three “high-load periods” in each half, separated by “lower-load periods”. The player load patterns were relatively similar between positions and occurred at approximately the same time points during the majority of matches”) as well as in the “Player load patterns and pacing”-section (“The present results show distinct player load patterns with three “high-load periods” in each half of the match (Fig 1), separated by “lower-load periods”, in most of the matches (Fig 2), …”).

To further aid in the visualization of the pattern for the reader, we have added references to Figure 1 and 2 where appropriate.

# Comment 25: L295 – “Locomotor activities” is a term too vague because it can include many different activities at low and high intensity. I suggest that you clear specifically what you mean regarding the cited studies [9,21,22].

# Response 25: We agree. This has now been changed to: “…pattern of HiR and sprint distance across studies…”

# Comment 26: L301-304 - Is this sentence referring to the studies 6 and 7? It is not clear.

# Response 26: We apologize for the lack of clarity. This sentence refers to our study, and has now been changed to: “In the present results, the patterns of the different high-intensity locomotor activities throughout match time show heterogeneity; patterns of sprint and HiR…”

Conclusions

# Comment 27: What are the practical applications for soccer coaches, staff members, or soccer science?

For example, can you provide some recommendations for soccer training?

# Response 27: Please see Response 11 to Reviewer #1.

# Comment 28: Also, you must point some limitations of this study.

# Response 28: A limitations section is now included.

---

## [Decision Letter · Decision Letter 1]

26 Aug 2020

PONE-D-20-16658R1

Player load in male elite soccer: comparisons of patterns between matches and positions

PLOS ONE

Dear Dr. Dalen,

Thank you for submitting your manuscript to PLOS ONE. After careful consideration, we feel that it has merit but does not fully meet PLOS ONE’s publication criteria as it currently stands. Therefore, we invite you to submit a revised version of the manuscript that addresses the small points raised during the review process by Reviewer 2. Please submit your revised manuscript by Oct 10 2020 11:59PM. If you will need more time than this to complete your revisions, please reply to this message or contact the journal office at plosone@plos.org. Please include the following items when submitting your revised manuscript:

We look forward to receiving your revised manuscript.

Kind regards,

Laurent Mourot

Academic Editor

PLOS ONE

Reviewers' comments:

Reviewer's Responses to Questions

**Comments to the Author**

1. If the authors have adequately addressed your comments raised in a previous round of review and you feel that this manuscript is now acceptable for publication, you may indicate that here to bypass the “Comments to the Author” section, enter your conflict of interest statement in the “Confidential to Editor” section, and submit your "Accept" recommendation.

Reviewer #1: All comments have been addressed

Reviewer #2: All comments have been addressed

2. Is the manuscript technically sound, and do the data support the conclusions?

Reviewer #1: Yes

Reviewer #2: Yes

3. Has the statistical analysis been performed appropriately and rigorously? 

Reviewer #1: Yes

Reviewer #2: Yes

4. Have the authors made all data underlying the findings in their manuscript fully available?

Reviewer #1: Yes

Reviewer #2: Yes

5. Is the manuscript presented in an intelligible fashion and written in standard English?

Reviewer #1: Yes

Reviewer #2: Yes

6. Review Comments to the Author

Reviewer #1: (No Response)

Reviewer #2: The paper "Player load in male elite soccer: comparisons of patterns between matches and positions” is a good contribution for the current state of the art in this specific topic.

Previously, I recommended to change your title and you in fact provide a changed title. However, I understand your answer to this topic, I still does not share the same idea because I really consider that player load and the other variables as well are equally relevant for the study. But I will let to your consideration.

Thus, I would like to suggest to change the first line sentence of the abstract to “Our primary aim was to explore the development of player load throughout match time (i.e., the pattern) using moving 5-min windows in an elite soccer team and our secondary aim was to compare player load patterns between different positions within the same team.”

This study highlights more knowledge on training load quantification methods that, per se, are very useful in different sports, physical activities and/or exercise training programs. They allow a better training control for different athletes or non-athletes. Therefore, the authors should be commended for their hard work in what appears to be an extensive study. Now, the current form of the manuscript provides some limitations and practical applications sections in this field that can be applied in other similar studies or other contexts.

Now, I would like to congratulate the authors for this revised version of the manuscript, as I now recommend it to be accept. Congratulations!

7. PLOS authors have the option to publish the peer review history of their article (what does this mean?). If published, this will include your full peer review and any attached files.

Reviewer #1: No

Reviewer #2: **Yes: **Rafael Franco Soares Oliveira

---

## [Author Response · Author response to Decision Letter 1]

27 Aug 2020

Dear Editor:

We thank you and the reviewers for helping us get our paper better and do hope you find the present version of the manuscript acceptable for publication. 

Comment 1, Reviewer #2: The paper "Player load in male elite soccer: comparisons of patterns between matches and positions” is a good contribution for the current state of the art in this specific topic.

Previously, I recommended to change your title and you in fact provide a changed title. However, I understand your answer to this topic, I still does not share the same idea because I really consider that player load and the other variables as well are equally relevant for the study. But I will let to your consideration.

Response 1: Thank you for accepting our view on this, we appreciate the understanding.

Comment 2, Reviewer #2: Thus, I would like to suggest to change the first line sentence of the abstract to “Our primary aim was to explore the development of player load throughout match time (i.e., the pattern) using moving 5-min windows in an elite soccer team and our secondary aim was to compare player load patterns between different positions within the same team.”

Response 2: We agree, and have made the change you request. The missing “was” was initially removed as a grammatical move to reduce words where we could, since we added more results to the abstract. To remain under the 300 word limit, we have now changed “which was also present in the majority of individual matches” to “which also occurred in the majority of individual matches”.

---

## [Editor Report · Decision Letter 2]

1 Sep 2020

Player load in male elite soccer: comparisons of patterns between matches and positions

PONE-D-20-16658R2

Dear Dr. Dalen,

We’re pleased to inform you that your manuscript has been judged scientifically suitable for publication and will be formally accepted for publication once it meets all outstanding technical requirements.

Kind regards,

Laurent Mourot

Academic Editor

PLOS ONE
---

## [Editor Report · Acceptance letter]

11 Sep 2020

PONE-D-20-16658R2 

Player load in male elite soccer: comparisons of patterns between matches and positions 

Dear Dr. Dalen:

I'm pleased to inform you that your manuscript has been deemed suitable for publication in PLOS ONE. Congratulations! Your manuscript is now with our production department. 

Kind regards, 

on behalf of

Dr Laurent Mourot 

Academic Editor

PLOS ONE